# In-situ airborne measurements of atmospheric and sea surface parameters related to offshore wind parks in the German Bight

Astrid Lampert[1], Konrad Bärfuss[1], Andreas Platis[2], Simon Siedersleben[3], Bughsin Djath[4],
Beatriz Cañadillas[5], Robert Hunger[1], Rudolf Hankers[1], Mark Bitter[1], Thomas Feuerle[1], Helmut Schulz[1],
Thomas Rausch[1], Maik Angermann[1], Alexander Schwithal[1], Jens Bange[2],
Johannes Schulz-Stellenfleth[4], Thomas Neumann[5], and Stefan Emeis[3]

[1]Institute of Flight Guidance, Technische Universität Braunschweig, Braunschweig, Germany
[2]Eberhard Karls University, Tübingen, Germany
[3]Karlsruhe Institute of Technology, Garmisch-Partenkirchen, Germany
[4]Helmholtz-Zentrum Geesthacht, Institute of Coastal Research, Geesthacht, Germany
[5]UL International, Oldenburg, Germany

**Correspondence:** Astrid Lampert (Astrid.Lampert@tu-braunschweig.de)

**Abstract.** Between 6 September 2016 and 15 October 2017, meteorological measurement flights were conducted above the German Bight in the framework of the project WIPAFF (Wind Park Far Field). The scope of the measurements was to study long-range wakes with extent larger than 10 km behind entire wind parks, and to investigate the interaction of wind parks and the marine atmospheric boundary layer. The research aircraft Dornier 128 of TU Braunschweig performed in total 41 measurement flights during different seasons and different stability conditions. The instrumentation consisted of a nose boom with sensors for measuring the wind vector, temperature and humidity, and additionally sensors for characterizing the water surface, a surface temperature sensor, a laser scanner, and two cameras in the visible and infrared wavelength range. A detailed overview of the aircraft, sensors, data post processing and flight patterns is provided here. Further, averaged profiles of atmospheric parameters illustrate the range of conditions. The potential use of the data set has been shown already by first publications. The data are publicly available in the world data center PANGAEA (https://doi.org/10.1594/PANGAEA.902845, Bärfuss et al., 2019a).

## 1  Introduction

The growing demand in renewable energy has led to large-scale installations of wind parks in the German Bight in the last decades. Before the project WIPAFF, satellite images of synthetic aperture radar (SAR) indicated modifications of the sea surface up to several 10 km downstream of wind parks (Christiansen and Hasager, 2005; Li and Lehner, 2013). Also numerical simulations suggested the existence of far-reaching wake areas with reduced wind speed and enhanced turbulence (Fitch et al., 2012). Motivated by these results, the lack of in-situ evidence and the need of understanding the wake effects as well as to collect in-situ data to validate existing wakes models, the project WIPAFF (Wind Park Far Field) was funded by the German Federal Ministry for Economic Affairs and Energy (BMWi). To verify these indirect estimates and quantify the effects of wind parks on the marine atmospheric boundary layer, flight measurements were performed. Within the project WIPAFF,

the flights were embedded in further stationary measurements of wind profiles by wind lidar systems and meteorological tower measurements at the masts FINO1 and FINO3. Satellite images provided statistical information on wake occurrence and extensions. WRF simulations were performed for some measurement flights, and verified by the observations. First results of the measurements have been presented in Platis et al. (2018), and WRF simulations and validation with observational data have been presented in Siedersleben et al. (2018a, b, 2019). The importance of atmospheric stability for the development of wakes has been addressed by Platis et al. (2019b) and will be investigated in detail in another publication. Analyses of wake recovery as a function of atmospheric stability and its representation in engineering model has been addressed by Cañadillas et al. (2020). The validation of an analytical model has been submitted by Platis et al. (2019a). An intercomparison of the significant wave height obtained by the airborne laser scanner and a wave model has been submitted by Bärfuss et al. (2019b). Analyses of large-scale wakes as a function of atmospheric stability by satellite remote sensing are published in Djath et al. (2018). For describing the surface roughness, airborne laser scanner data have been used to fill the gap between buoy and satellite observations, as well as for validation of wave simulations in the German Bight (Bärfuss et al., 2019b). An overview publication of the main results of the project WIPAFF has been submitted (Platis et al., 2019b).

## 2 Research aircraft Dornier-128

The airborne measurements were performed with the research aircraft Dornier 128-6 with call sign D-IBUF. The Dornier 128-6 is a twin-engined turboprop powered research aircraft used for different research fields (Fig. 1). Besides all necessary avionic instrumentation for flights especially at very low altitudes it has versatile sensor equipment combined with a powerful data acquisition system. The Dornier 128-6 has been used for different kind of meteorological research, in particular for investigations of processes in the atmospheric boundary layer (ABL):

Several measurement campaigns were performed above the North Polar Ocean, the Golf of Bothnia and in particular over the sea ice edge with the aim to investigate meteorological processes at the intersection of sea ice and open water for off-ice flow, resulting in convective cells and cloud streets (Brümmer et al., 1994, 2002). The aircraft was used for studying the sources and chemical cycle of anthropogenic ozone in the ABL (Corsmeier et al., 2002). The impact of inhomogeneous terrain on the turbulent exchange processes between ground and the atmospheric boundary layer was investigated by Bange et al. (2002, 2006). Convective processes up to the formation of thunderstorms in the atmosphere were studied by direct measurements in the clouds and drop sondes released from the aircraft (Groenemeijer et al., 2009). The aircraft was deployed above the Mediterranean to investigate cyclones and mesoscale convective systems to understand which atmospheric conditions lead to devastating thunderstorms, and to improve the forecasting of such systems (Drobinski et al., 2013; Ducrocq et al., 2013; Sodemann et al., 2017). A comparison of convective boundary layer conditions between wind lidar and airborne measurements was performed (Adler et al., 2019). The aircraft was used for an intercomparison of fast humidity sensors (Lampert et al., 2018). There is an online graphical display with time series and vertical profiles of all important measured and calculated meteorological parameters (wind speed, wind direction, turbulent kinetic energy, eddy dissipation rate, temperature, potential temperature, humidity, surface temperature), enabling the onboard scientist to modify the ongoing mission based on the measured param-

eters, if necessary. Like this, it was possible to adapt the flight pattern during the mission. For the flights at the wind parks at
low altitudes down to 60 m, a special permission was required.

In the data set, the aircraft position, altitude, velocity in all three directions, the pitch and roll angle and heading are provided,
which are necessary for the data analyses. Further, the radar altitude above ground is given.

## 3  Sensors and data processing

For meteorological flight campaigns, the Dornier-128 aircraft can be equipped with a nose boom and additional sensors.
The sensor system is specialized for meteorological measurements (Corsmeier et al., 2001); the central sensor package is
contained in the nose boom (Fig. 2). This concentration of meteorological sensors with high temporal resolution for measuring
temperature, humidity, wind speed and wind direction leads to a high spatial resolution of the data. With a mean ground
speed of about $65 \, \mathrm{m \, s^{-1}}$ and a measuring rate of $100 \, \mathrm{Hz}$ the spatial resolution of the measurements is higher than $1 \, \mathrm{m}$. The
application of the nose boom for measuring the wind vector, temperature and humidity, and the surface temperature sensor
represent standard research components of the Dornier-128. The laser scanner for sea surface deflection and nadir looking
cameras were integrated specifically for the WIPAFF campaigns. In the following, the sensors and the standard calibration
procedures are presented.

### 3.1  Temperature

Temperature measurements are performed by two complementary sensors, the slow, but highly accurate 102DB1AG tempera-
ture sensor (Rosemount, US) with an accuracy of $\pm 0.1 \, \mathrm{K}$, and the 102E4AL sensor (Rosemount, US) with a fast response time
and an accuracy of $\pm 0.25 \, \mathrm{K}$ plus 0.5% of the temperature to be measured in $^\circ \mathrm{C}$. The slow sensor is heatable, but heating was
not switched on, as no icing conditions were present during the flights.

The total temperature $\mathrm{T}_{total}$ is derived by applying a recovery factor to the raw measurements to compensate the self heating
effect. From the total temperature in K, the static temperature $\mathrm{T}_{stat}$ in K is derived adiabatically (Stickney et al., 1994):

$$T_{stat} = T_{total} \cdot \left( \frac{p_{stat}}{p_{total}} \right)^{\frac{\kappa - 1}{\kappa}} \tag{1}$$

Here, $\mathrm{p}_{stat}$ is the static pressure and $\mathrm{p}_{total}$ is the total pressure, the sum of static and dynamic pressure. $\kappa$ is the heat capacity
ratio with a value of 1.4.

The calibration of the temperature sensors is done by applying specific resistance values corresponding to specific temperatures
as stated by the manufacturer.
In the PANGAEA data set Bärfuss et al. (2019a), the static air temperature derived from the fast sensor is provided, after using
the slow sensor for quality check. The parameter is simply called "air temperature".

## 3.2 Humidity

For measuring humidity, three different measurement principles are used: A capacitive Vaisala Humicap HMP233, Finland, a dew point mirror TP 3-S of Meteolabor, Switzerland, and a Lyman-Alpha optical sensor L-6 / HMS-2 of Buck Research, US. The humidity sensors have a joint heatable inlet, and other parameters like temperature and pressure are recorded for the humidity channel as well. The humidity sensors are cleaned and calibrated before each meteorological measurement campaign by applying saturated salt solutions with known relative humidity in an equilibrium state. In the PANGAEA data set, the relative humidity of the dew point mirror is provided as a reference with good accuracy of the absolute values (accuracy of the dew point specified by the manufacturer as 0.15 K) The temporal resolution is composed of a time <0.5 s for the condensation process or temperatures above 0°C, plus a time delay proportional to the magnitude of abrupt changes in the dew point (5 K s$^{-1}$). The relative humidity of the Lyman-Alpha sensor with much shorter response time is provided for deriving fluctuations.

## 3.3 Pressure and wind

With the 5-hole probe of Rosemount, US, and pressure transducers of Setra, US, the static and dynamic pressure, as well as the air flow angles are retrieved in the aircraft-fixed coordinate system. All inertial data (position, ground speed, Eulerian angles) were derived from the complementary use of the inertial measurement platform iNAV-RQH-1003 of iMAR, Germany, operated in parallel to the former standard system Lasernav of Honeywell, US, and a NovAtel GPS OEM-6, Canada. These input parameters are then used to calculate the wind vector according to the formulation in Lenschow (1972), whereas the fundamental vector difference equation in geodetic coordinates is

$$\overline{V}_{w_g} = \overline{V}_{K_g} - \overline{V}_g \tag{2}$$

$\overline{V}_{w_g}$ denotes the wind vector, $\overline{V}_{K_g}$ the flight path velocity, and $\overline{V}_g$ the velocity vector of the aircraft with respect to the air. In the PANGAEA data set, all three components of the wind vector are provided at 100 Hz resolution to derive horizontal wind speed, wind direction and turbulent properties. Further, the air density derived from the static pressure and temperature is given.

## 3.4 Sea surface temperature

An infrared KT15.82D sensor of Heimann, now Heitronics, Germany, is used to determine the surface temperature. It has an accuracy of $\pm 1.2$ K at 20°C surface temperature and a temporal resolution of 20 Hz. If no clouds are between the sensor and the surface, the surface temperature measurements are not influenced by the atmospheric temperature or humidity distribution. The footprint size is 10 m at a distance of 900 m for the specific system. In the PANGAEA data set, the parameter is called surface temperature.

## 3.5 Sea surface deflection

The scanning laser system VZ-1000 of Riegl, Austria, is deployed to record the relative sea surface deflection and to derive parameters like the significant wave height.

From the point measurements in the scanner's coordinate system $\begin{pmatrix} v_{x_{body}} \\ v_{y_{body}} \\ v_{z_{body}} \end{pmatrix}$, aircraft attitude corrections using Eulerian angles $\Psi, \Theta, \Phi$ are applied to rotate aircraft body fixed coordinates into the geodetic coordinate system (positive directions

East, North, Up), which then are geolocated by applying the aircraft position $\begin{pmatrix} p_{x_{body}} \\ p_{y_{body}} \\ p_{z_{body}} \end{pmatrix}$ in the following manner:

$$\begin{pmatrix} p_{x_{geo}} \\ p_{y_{geo}} \\ p_{z_{geo}} \end{pmatrix} = R^{body}_{geo}(-\Psi, -\Theta, -\Phi) \begin{pmatrix} v_{x_{body}} \\ v_{y_{body}} \\ v_{z_{body}} \end{pmatrix} + \begin{pmatrix} p_{x_{body}} \\ p_{y_{body}} \\ p_{z_{body}} \end{pmatrix} \qquad (3)$$

Subsequently, the surface deflection $\eta$ is calculated out of the georeferenced point cloud using mean sea level. The system's effective rate of the distance measurements is up to 122 kHz, but decreases over water because of specular reflections. Accuracy and resolution along the beam direction are stated to be less than 10 mm, and measurements can be taken up to a distance of

450 m with the settings used during flight campaigns.

Significant wave height (SWH) $H_s$, defined in the spatial domain, is used to describe the sea surface. The approximation $H_s \approx H_{m_0} = 4 \cdot \sigma_\eta$ mentioned in Young (1999) was used, since this simple calculus only depends on the standard deviation $\sigma_\eta$ in sea surface deflection. Here, $H_{m_0}$ is four times the standard deviation of the sea surface deflection measurements, normally defined in the frequency domain. In this case, it is derived from measurements in the space domain. With a deflection

measurement rate of more than 3 kHz over water, this produces stable results in SWH estimation. As scan pattern, a line scan pattern rectangular to the flight direction was used, which provides a spatial resolution of about 0.5 m x 0.5 m between measurement points perpendicular to and along the flight trajectory. In the PANGAEA data set sea surface deflections $\eta$ are analysed for standard deviation within a time window of 10 s.

## 3.6 Cameras

Two downward looking cameras, one for the visible wavelength range (MV1-D1312-G2 of Photonfocus, Switzerland), and one for the infrared range (A35SC of FLIR, Germany) were deployed in the fuselage to document the sea surface. The images are influenced by sun glint and by varying cloud cover. The exposure time of the visible camera was adapted manually. The retrieval of specific parameters requires additional intensive processing of the images.

The large data sets are available at the Institute of Flight Guidance upon request. They are not included in the data base. Further,

handheld cameras were used to document the overall impression, clouds and special features. They are not included in the data base either.

## 4  Flight planning and flight patterns

In preparation of the measurement campaign, flight patterns were programmed to systematically probe the far field wakes behind the wind park clusters including Godewind and Amrumbank West (named N-3 and N-4 according the offshore areas
specified by the German Federal Hydrographic Agency) for different wind directions every 10°. Altogether, 41 measurement flights were conducted during different seasons, wind direction, wind speed and stability. An overview of the flights performed during WIPAFF and meteorological conditions is shown in Tab. 1. A map with all flight paths flown during WIPAFF is provided in Fig. 3. During the flights, no instrument failures occurred. Only during one flight, the data acquisition had to be re-started (Flight 35).
Generally, flights were performed downwind of Amrumbank West for a wind direction sector of 80° to 200°, and downwind of Godewind for a sector from 160° to 350°. However, there are exceptions for particular reasons (e.g. during Flight 5 for consecutively probing the wakes of both wind parks, Flight 6 for investigating the changes of the wind field above the wind park). Depending on the wind direction, either the cluster of Amrumbank West or Godewind was investigated, as flights were only performed above water and only above German controlled air space. The flight patterns were prepared with a
software developed at the Institute of Flight Guidance. A special function is implemented for programming meander patterns automatically after defining a starting point and the length and distance of legs. The flights were performed from the airports Wilhelmshaven (ICAO code EDWI), Husum (EDXJ) or Borkum (EDWR) depending on wind direction and runway orientation, and proximity to the wind parks.

### 4.1  Meander at hub height (MEANDER)

To quantify the wakes behind offshore wind parks and determine the wake length, meander flight patterns at hub height perpendicular to the prevailing wind direction were applied (MEANDER). An example is provided in Fig. 4. For these flight patterns, isolated wind parks with long distance of unobstructed water surface downwind were selected. The flight pattern typically started with a leg 500 m downstream of the last wind turbines. The distance to the next flight legs was set as 10 km. The flight altitude was adapted to the hub height and was either 90 m (Amrumbank West) or 120 m (Godewind). On the way
to the wind park and after the meander pattern, vertical soundings from 60 to 550 m, sometimes up to 1000 m were performed to investigate atmospheric stability. For unstable conditions, the distance between the flight legs perpendicular to the wind direction was shortened. This flight pattern was used for 26 out of the 41 flights.

Results of meander flight patterns have been published in Platis et al. (2018); Siedersleben et al. (2018a); Cañadillas et al. (2020).

## 4.2 Vertical cross sections (CROSS)

To quantify the vertical extent of wakes, several cross sections at the same distance downwind of the windpark were flown perpendicular to the wake at different altitudes (CROSS). Typical flight altitudes were 60 m, 90 m, 130 m, 150 m and 200 m. On the way to the wind park and back, vertical soundings from 60 to 550 m, sometimes up to 1000 m were performed to investigate atmospheric stability. Such flight patterns were applied during 8 out of the 41 measurement flights.

Results of the cross section patterns measured on 10 September 2016 have been published in Siedersleben et al. (2018a, b).

## 4.3 Above wind parks (ABOVE)

To quantify the interaction of the wind parks and the atmospheric boundary layer, a flight pattern with legs upwind, above and downwind the wind park was repeated several times (ABOVE, Fig. 5). Individual flight legs had a length of 45 km. The flight pattern was flown at an altitude of 65 m above the top of the rotor blades. On the way to and from the wind park, vertical soundings were performed to obtain information on atmospheric stability.

Such flight patterns were performed during 18 out of the 41 measurement flights.

Results of the flight patterns above wind parks have been published by Siedersleben et al. (2019).

## 5 Atmospheric conditions

The low-level flights were conducted under visual flight conditions. A minimum visibility of 10 km, a minimum cloud ceiling of 1.000 ft and no precipitation were required. Therefore the results are not statistically representative of atmospheric conditions above the North Sea. On the contrary, in particular in spring 2017, flights were only possible on occasional days.

During the flights, a large variety of atmospheric conditions was encountered. The focus of the flight was on far reaching wakes, therefore days with very stable atmospheric conditions were the preferred option. However, for comparison, days with less stable conditions were probed as well. In the following, vertical profiles of the mean and extreme values of temperature, potential temperature, wind speed and water vapour mixing ratio are presented. During each flight, different vertical profiles were obtained. First, a mean profile for each flight was calculated. Then all 41 profiles from the 41 flights were averaged again. For each height, the minimum and maximum values were determined from the 41 profiles representing each one particular flight.

### 5.1 Temperature

The temperatures encountered during the WIPAFF flights span a broad range, as the flights were performed during different seasons. The near-surface air temperature varied between 7° C and 25° C (see Fig. 6). Overall, the temperature decreased with altitude. Below 60 m, data are only available during take-off and landing. Therefore, the temperature inversion below 60 m is not a typical feature above the North Sea, and therefore provided as dotted line. The averaged and maximum temperature

profiles show a sudden decrease at an altitude of around 500 m and around 950 m. This is probably an artifact from the averaging method, and it is not visible in individual temperature profiles.

## 5.2 Stability

Atmospheric stability is strongly related to season and wind direction. In spring and summer, the water surface warms relatively slowly, whereas atmospheric temperatures above land are subject to a strong diurnal cycle. Therefore, flow from land to sea during day very frequently results in stable conditions. For northerly wind directions, the air temperature is typically similar to the water surface temperature, so unstable or neutral conditions prevail. A rough overview of all conditions is indicated in Fig. 7. In the mean profile of the potential temperature, a clear increase is observed for the altitude interval 60 to 100 m. Also up to the altitude of 200 m, in the range of the rotor blades, an overall small increase of potential temperature with height is observed. The decrease of average and maximum potential temperatures with height at around 500 m and 950 m are probably artifacts form the averaging method and are not visible in the profiles of individual flights.

As stability typically changes with distance to the coast and with the diurnal cycle, the categorization of the flights according to one specific stability parameter is difficult and needs thorough discussion, which is addressed in Platis et al. (2019b) and will be subject to another publication. The exact altitude of the temperature inversion in relation to the rotor geometry plays a crucial role for the modification of temperature and humidity profiles in the wake areas (Siedersleben et al., 2018b).

## 5.3 Wind speed

During the flights, wind speed at hub height varied between $2\,\mathrm{m\,s^{-1}}$ and $17\,\mathrm{m\,s^{-1}}$. The typical cut-in speed at which offshore wind turbines start producing power is around $3\,\mathrm{m\,s^{-1}}$. The rated speed for offshore wind turbines is typically designed as $12\,\mathrm{m\,s^{-1}}$, and the cut-out speed, where wind turbines are shut down is at $25\,\mathrm{m\,s^{-1}}$. The wind speed typically increases more strongly with altitude for stable conditions. An overview of all wind speed profiles encountered during the flights is shown in Fig. 8. The strong increase of wind speed from the surface to 50 m is an artifact as these altitudes were only sampled during take-off and landing.

## 5.4 Wind direction

Measurement flights were performed for mean wind directions at hub height between 80° and 330°. Wind directions from NW were always associated with unstable or neutral atmospheric stability. This wind sector was investigated mainly for comparison. The main focus was on stable conditions, and therefore wind from land.

## 5.5 Humidity

The profiles of humidity varied strongly depending on stability. For unstable conditions, an enhanced water vapour mixing ratio directly above the water surface was present. For stable conditions, humidity was often increased at higher altitudes, which in most cases is most likely caused by advection of air masses with higher water vapour mixing ratio. Depending on the altitude of

the temperature inversion in relation to the altitude of the rotor blades, humidity was either increased or decreased in the wake (Siedersleben et al., 2018b). As the relative humidity is temperature dependent, profiles of the water vapour mixing ratio are shown in Fig. 9. The water vapour mixing ratio varied between 2 and $14\,\mathrm{g\,kg^{-1}}$. For the mean profile, a sharp decrease of the mixing ratio with altitude is present at around 500 m. This corresponds to the altitude with a change of stability as indicated by the potential temperature and an increase in the wind speed, indicating the mean altitude of the marine atmospheric boundary layer.

For completeness and for interpretation of the data, the prevailing cloud conditions based on visual observations are indicated in Table 1 and 2.

## 6 Modification of wind field by the coast

The profiles of temperature and wind speed are modified by coastal effects (e.g. Dörenkämper et al. (2015); van der Laan et al. (2017)). During the WIPAFF campaign, climb and descent flights were performed on the way to the wind park measurement area and back. As an example of current and future research, the modification of the wind by coastal effects was investigated. Figure 10 shows the difference of the wind speed at hub height (120 m) for each profile minus the wind spee at hub height obtained during the profile closest to the coast. The fetch length is defined as the mean length that the air travelled above open water along the wind direction. Only flights are included where it was possible to determine the fetch length (not from North and West, as the distances to the next coast lines are too large). There is a large scatter in the data. Figure 11 shows the same data points. However, they are grouped by wind direction. There is still large scatter in the different data sets. However, depending on wind direction, the wind speed either increases or decreases with fetch length. This shows that there are more parameters required to explain the modification of wind speed besides the fetch length. A weakness of this analysis is that the profiles were not obtained along the mean wind direction. So air masses do not have the same origin, and besides the fetch length, variability along the coast line influences the results. More investigation is required to understand and parameterize the coastal effect.

## 7 Conclusions

The WIPAFF flights are the only available data set to date, from which the impact of long-range wakes can be derived systematically and independent of infrastructural constraints like the location of masts. Under stationary conditions, the aircraft data provide detailed information on the modifications of the flow field downstream of wind parks. For the interpretation, spatial changes in the flow field caused by synoptic-scale differences have to be taken into account, e.g. North-South gradients in wind speed and wind direction. Also temporal changes of the wind field have to be taken into account for the four-hour flights. During this time period, stationary conditions cannot always be assumed. For the data interpretation, short-time changes as frontal systems, synoptic-scale continuous changes, and modification of air masses and stability due to the diurnal cycle of solar radiation have to be considered.

255 The data set can be used complementary to other wind field observations by satellite, at the wind parks and for lidar measurements and to validate specific results, as suggested by Schneemann et al. (2019).

The unique data has been the base for different studies, proving for the first time directly the horizontal extension of wakes downwind of offshore wind parks (Platis et al., 2018), quantifying the wind speed recovery in dependence of stability (Cañadillas et al., 2020; Platis et al., 2019a), and for the validation of the WRF mesoscale model (Siedersleben et al., 2018a, b, 2019),

260 which can then be used for larger scales and future wind energy scenario calculations.

## 8 Data availability

The data are publicly available at https://doi.pangaea.de/10.1594/PANGAEA.902845 (Bärfuss et al., 2019a). Each data set of a flight in ascii format as tab-delimited text has a size of around 140 MB. The zip file containing all data sets as tab-delimited text has a size of around 750 MB. Upon request, additional laser scanner raw data, camera images in the visible and infrared

265 wavelength range and manual cloud photographs are available. Satellite images of Sentinel-1 (A and B) and are freely available at https://scihub.copernicus.eu/.

*Author contributions.* AL and KB wrote the text with contributions from all co-authors. KB and RH created the figures. All authors helped to design the flight patterns and analysed the data. AL, KB, RH, MB, TF, HS, TR, AP and JB realized the measurement campaigns. AS provided the software for easily programming the flight patterns. TN and SE developed the project idea.

270 *Competing interests.* The authors declare that they have no conflict of interest.

*Acknowledgements.* The airborne observations have been funded by the project WIPAFF of the German Federal Ministry for Economic Affairs and Energy under grant number 0325783B. The authors would like to thank Christof Lüpkes and an anonymous referee for their valuable comments. The authors would like to thank Martin Dörenkämper and Jörge Schneemann for providing the correct coordinates of the wind turbines.

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

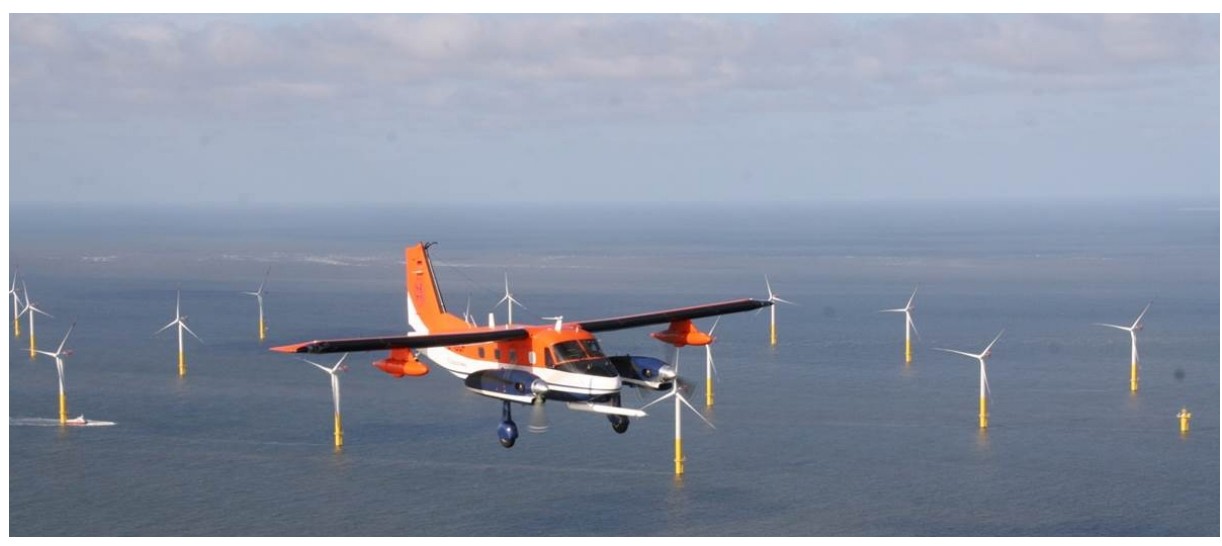

**Figure 1.** The Dornier 128-6 D-IBUF in front of a wind park on 10 August 2017. Photo: Mark Bitter, TU Braunschweig

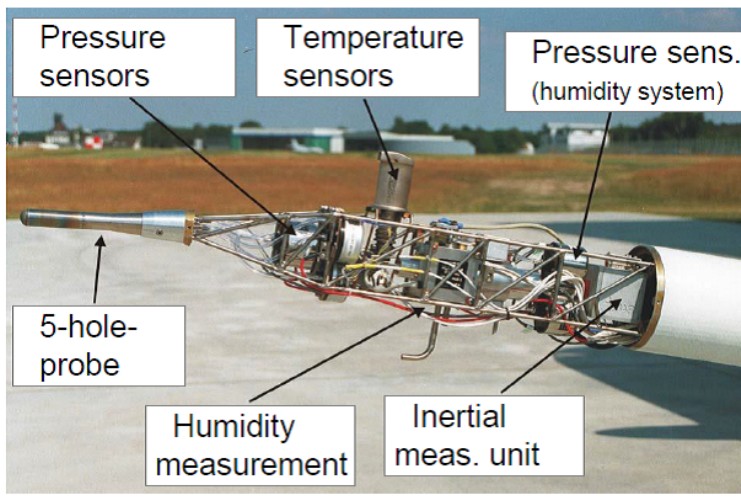

**Figure 2.** Nose boom and instrumentation of the research aircraft Dornier 128-6 D-IBUF. Photo: Mark Bitter, TU Braunschweig

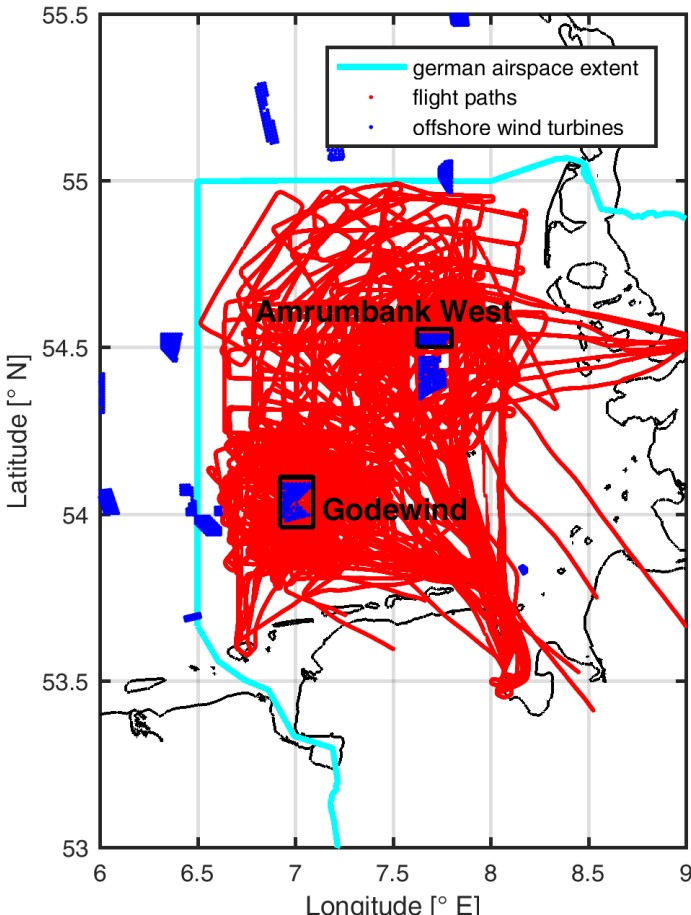

**Figure 3.** Tracks of all measurement flights performed during the WIPAFF experiment. The flight tracks are indicated in red. Each wind turbine installed until 2017 is represented by a blue dot. The measurements were performed from the airports Wilhelmshaven, Husum and Borkum. The wind parks Godewind and Amrumbank West are indicated with a black box. Flight tracks end when the data acquisition was shut down. The extent of the German air space is indicated by a light blue line.

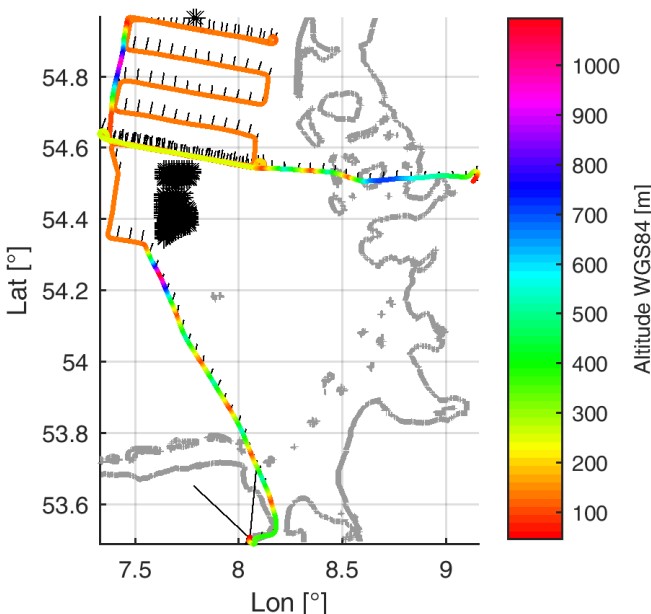

**Figure 4.** Example of the meander pattern to investigate the wake behind a wind park. Flight 7 was performed on 10 September 2016. It started at Wilhelmshaven airport and ended at Emden airport for refuelling. The black stars indicate individual wind turbines. The colours show the flight altitude. On the way to the wind park and back, vertical climbs and descents were performed to study atmospheric stability. The wind barbs indicate wind direction and a first idea of wind speed, which is proportional to the length of the wind barbs.

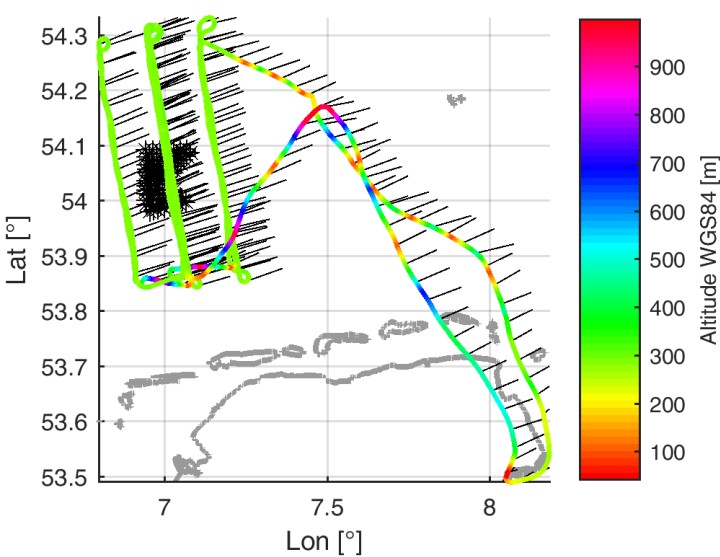

**Figure 5.** Example of the pattern above the wind park to investigate the downward mixing. Flight 39 was performed on 14 September 2017. It started and ended at Wilhelmshaven airport. The black stars indicate individual wind turbines. The colours show the flight altitude. On the way to the wind park and back, vertical climbs and descents were performed to study atmospheric stability.

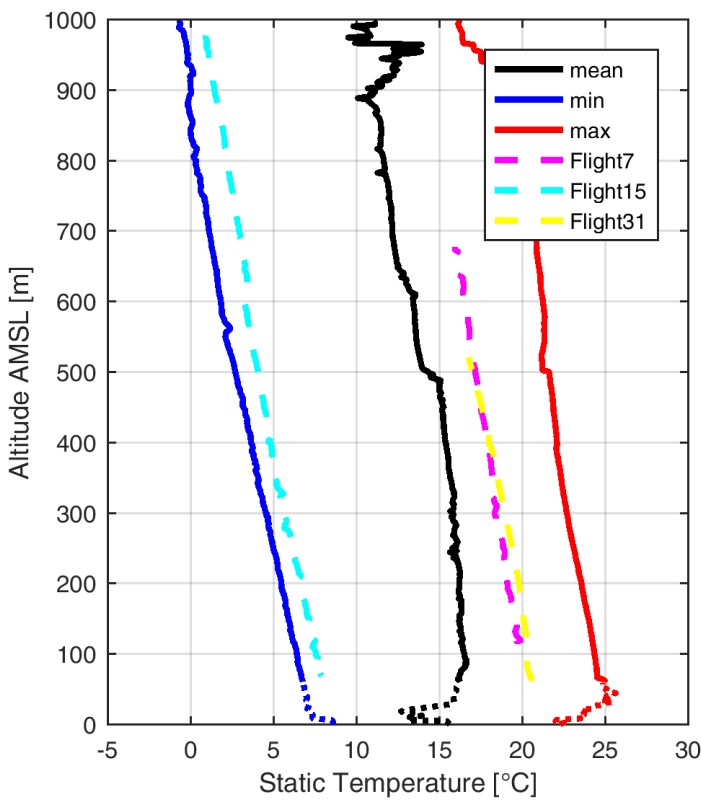

**Figure 6.** Average temperature profile (black) and range of temperatures (minimum blue, maximum red) encountered during the 41 WIPAFF measurement flights. Additionally, the averaged profiles of Flight 7 (magenta), Flight 15 (cyan) and Flight 31 (yellow) are included. As altitudes below 60 m altitude were only performed during take-off and landing, i.e. above land and not above the North Sea, the profiles are provided in dotted lines.

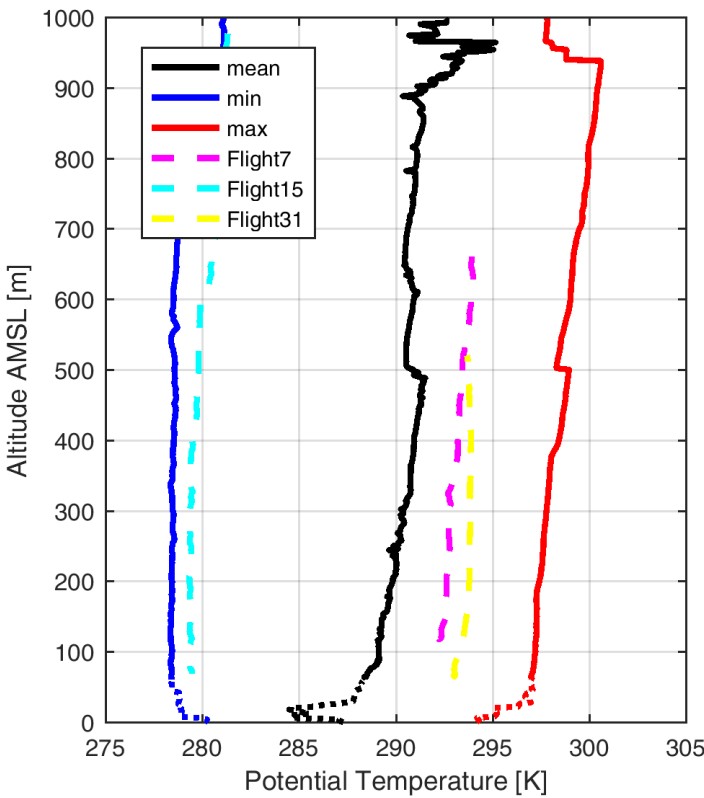

**Figure 7.** Average stability conditions (black) and range of stability (minimum blue, maximum red) encountered during the 41 WIPAFF measurement flights. Additionally, the averaged profiles of Flight 7 (magenta), Flight 15 (cyan) and Flight 31 (yellow) are included. As altitudes below 60 m altitude were only performed during take-off and landing, i.e. above land and not above the North Sea, the profiles are provided in dotted lines.

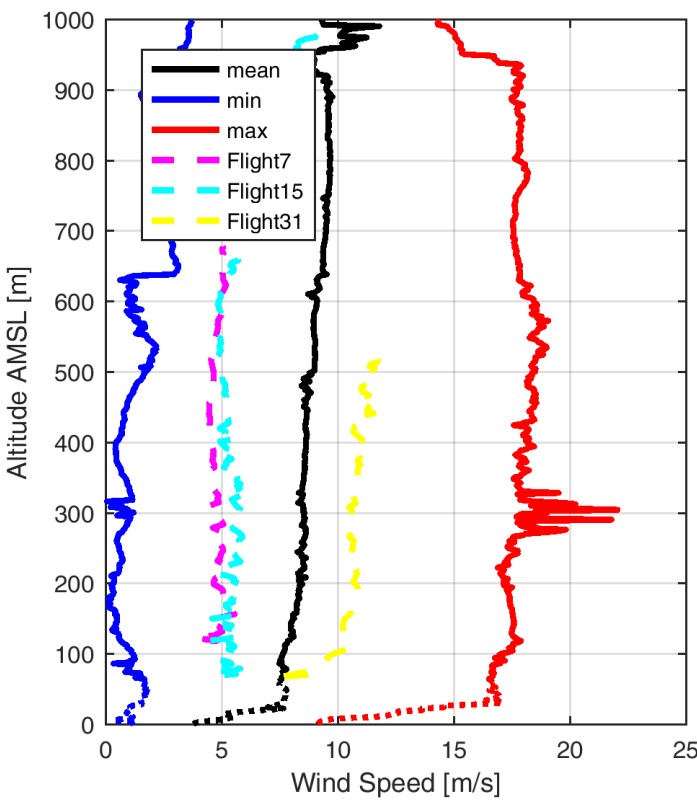

**Figure 8.** Average wind speed profile (black) and range of wind speed (minimum blue, maximum red) encountered during the 41 WIPAFF measurement flights. Additionally, the averaged profiles of Flight 7 (magenta), Flight 15 (cyan) and Flight 31 (yellow) are included. As altitudes below 60 m altitude were only performed during take-off and landing, i.e. above land and not above the North Sea, the profiles are provided in dotted lines.

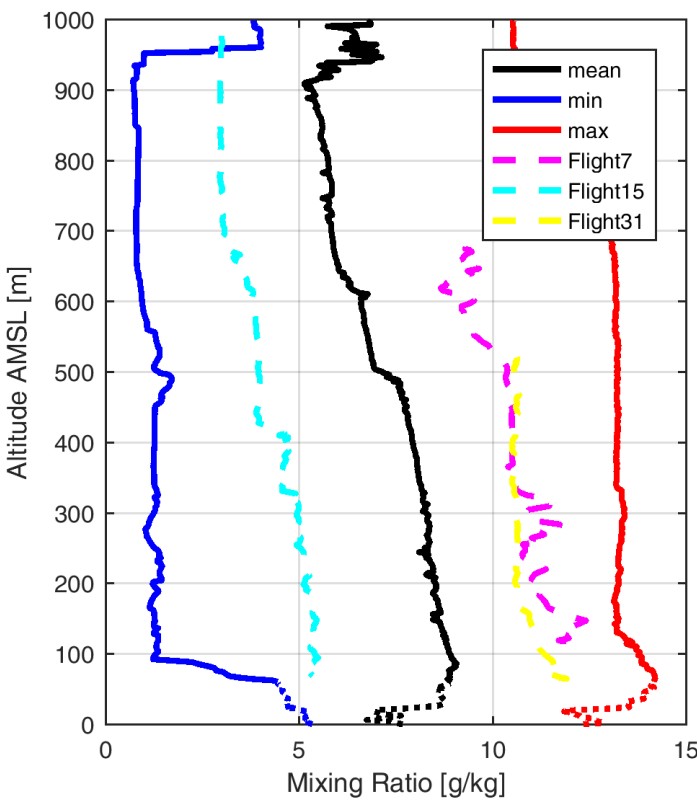

**Figure 9.** Average profile of water vapour mixing ratio (black) and range of mixing ratio (minimum blue, maximum red) encountered during the 41 WIPAFF measurement flights. Additionally, the averaged profiles of Flight 7 (magenta), Flight 15 (cyan) and Flight 31 (yellow) are included. As altitudes below 60 m altitude were only performed during take-off and landing, i.e. above land and not above the North Sea, the profiles are provided in dotted lines.

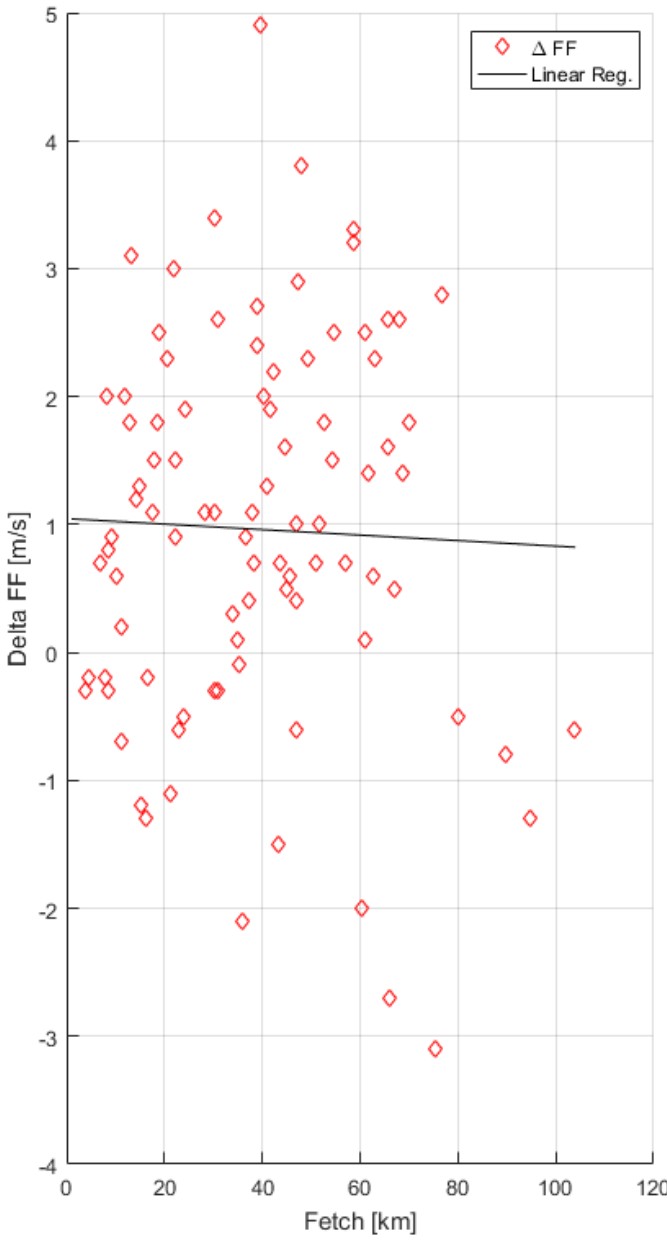

**Figure 10.** Changes of the wind speed at hub height (120 m) from the profile closest to the coast line to the other vertical profiles depending on fetch length. The vertical profiles of Flight 1, 3, 4, 5, 6, 7, 8, 9, 10, 13, 18, 19, 24, 30, 31, 32, 35, 36, 37, 38, 40, and 41 are included. Excluded were flights with wind direction from North or West, where no fetch length can be determined.

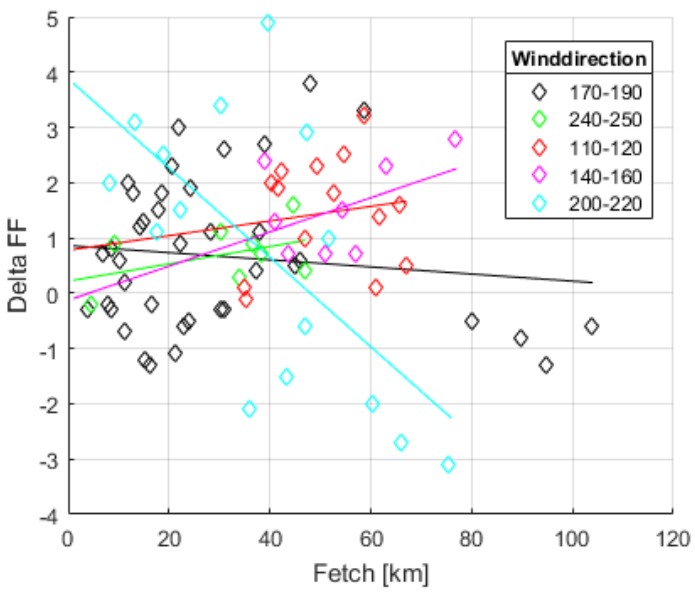

**Figure 11.** Changes of the wind speed at hub height as in Fig. 10. Different sectors of the wind direction are indicated in different colours.

**Table 1.** Overview of the WIPAFF measurement flights 1–20. The flight patterns are MEANDER(M), CROSS (C) or ABOVE (A) as indicated in Sect. 4. The wind parks are Godewind (GD) or Amrumbank West (AM). Information on cloud conditions is not always available (n.a.). Sentinel 1A and 1B satellite overpasses on the same day are indicated as well.

| flight number | date | flight time [UTC] take off – landing | wind park | flight pattern | main flight altitude [m] | wind speed [m s$^{-1}$] | wind dir [°] | cloud conditions | satellite [UTC] |
|---|---|---|---|---|---|---|---|---|---|
| 1 | 6 Sep 2016 | 12:13-15:20 | AM | M | 90 | 7 | 190 | n.a. | - |
| 2 | 7 Sep 2016 | 07:27-10:43 | AM | M, C | 90 | 4 | 210 | n.a. | - |
| 3 | 7 Sep 2016 | 12:06-13:59 | AM | C | 90, 120, 150 | 4 | 190 | n.a. | - |
| 4 | 8 Sep 2016 | 08:39-12:23 | AM | M, C | 60, 90, 120, 150 | 8 | 120 | Ci | 1A 17:09 |
| 5 | 9 Sep 2016 | 09:00-12:40 | GD, AM | M | 90, 120 | 6 | 240 | n.a. | - |
| 6 | 9 Sep 2016 | 13:42-17:10 | AM | A | 300 | 6 | 250 | Cu | |
| 7 | 10 Sep 2016 | 07:43-11:13 | AM | M,C | 60, 90, 120, 150, 200 | 7 | 190 | clear sky | 1A 05:41 |
| 8 | 10 Sep 2016 | 12:17-15:58 | AM | M,C | 60, 90, 120, 150, 200 | 4 | 190 | Ci | - |
| 9 | 30 Mar 2017 | 13:56-17:02 | GD | M | 120 | 15 | 240 | As, Ci | 1B 17:16 |
| 10 | 31 Mar 2017 | 13:36-16:59 | GD | A | 60, 90, 120, 200, 250 | 13 | 180 | As, Ci | - |
| 11 | 5 Apr 2017 | 13:42-16:33 | GD | M,A | 120, 200, 250 | 14 | 310 | Sc | 1A 17:17 |
| 12 | 6 Apr 2017 | 13:29-16:20 | GD | M,A | 120, 200, 250 | 10 | 310 | Sc, As | - |
| 13 | 9 Apr 2017 | 10:36-14:05 | GD | M,C | 60, 120, 200, 250 | 7 | 220 | clear sky | - |
| 14 | 9 Apr 2017 | 14:31-17:16 | GD | C | 60, 120, 200, 250, 350 | 4 | 200 | clear sky | - |
| 15 | 11 Apr 2017 | 09:15-13:09 | GD | A | 250, 300 | 8 | 280 | Cu, As | - |
| 16 | 11 Apr 2017 | 14:07-17:07 | GD | M | 120, 200 | 8 | 260 | St, showers | 1B 17:16 |
| 17 | 13 Apr 2017 | 11:23-14:40 | GD | M,A | 120, 250, 300 | 13 | 290 | Cu | 1B 05:48 |
| 18 | 17 May 2017 | 11:31-14:27 | AM | C | 90, 120, 150, 200, 250 | 8 | 110 | Sc | - |
| 19 | 17 May 2017 | 15:16-17:45 | AM | M | 220 | 12 | 120 | Ac | 1B 17:16 |
| 20 | 23 May 2017 | 07:53-10:41 | GD | M,A | 120, 250 | 6 | 250 | Ci, Ac, Sc | - |

**Table 2.** Overview of the WIPAFF measurement flights 21–41 containing the same information as in Table 1.

| flight number | date | flight time [UTC] take off – landing | wind park | flight pattern | main flight altitude [m] | wind speed [m s$^{-1}$] | wind dir [°] | cloud conditions | satellite [UTC] |
|---|---|---|---|---|---|---|---|---|---|
| 21 | 23 May 2017 | 11:18-15:10 | GD | M,A | 120, 250 | 11 | 310 | Cu | 1A 17:16 |
| 22 | 24 May 2017 | 05:40-09:33 | GD | M,A | 120, 240 | 8 | 300 | Sc | - |
| 23 | 24 May 2017 | 10:13-14:10 | GD | A | 250, 300 | 9 | 270 | St | - |
| 24 | 27 May 2017 | 08:45-11:56 | AM | M | 90 | 10 | 150 | clear sky | - |
| 25 | 27 May 2017 | 12:39-16:35 | AM | M | 90 | 12 | 140 | clear sky | - |
| 26 | 31 May 2017 | 09:04-11:45 | GD | A | 250 | 8 | 290 | Cu | 1B 05:48 |
| 27 | 31 May 2017 | 13:00-16:49 | GD | A | 250 | 9 | 290 | Ci, Cu | - |
| 28 | 1 Jun 2017 | 07:06-10:53 | AM | A | 90, 150, 210 | 6 | 300 | Cu | 1A 05:40 |
| 29 | 2 Jun 2017 | 06:47-10:39 | AM | M | 60, 90, 120, 150, 220 | 4 | 170 | few Ci | - |
| 30 | 8 Aug 2017 | 08:39-12:32 | AM | M | 90 | 10 | 80 | St | - |
| 31 | 8 Aug 2017 | 13:06-17:06 | AM | M | 90 | 14 | 80 | St | - |
| 32 | 9 Aug 2017 | 08:34-12:36 | AM | A | 90, 200 | 15 | 210 | St, rain showers | - |
| 33 | 9 Aug 2017 | 13:09-17:04 | AM | A | 90, 200 | 13 | 240 | Cu, rain showers | 1B 17:16 |
| 34 | 10 Aug 2017 | 10:49-14:53 | AM | A | 90, 200 | 5 | 340 | Ac | 1A 17:09 |
| 35 | 14 Aug 2017 | 10:08-14:07 | AM | M | 90 | 8 | 150 | Ci | - |
| 36 | 14 Aug 2017 | 14:40-18:30 | AM | M | 90 | 7 | 120 | Ci | - |
| 37 | 15 Aug 2017 | 07:22-11:15 | GD | M | 120 | 8 | 180 | Sc, Ac | 1A 17:17 |
| 38 | 17 Aug 2017 | 06:06-10:09 | AM | M | 90 | 11 | 160 | St | 1A 05:49 |
| 39 | 14 Oct 2017 | 12:59-16:40 | GD | A | 250 | 15 | 260 | St | 1A 17:17 |
| 40 | 15 Oct 2017 | 07:06-11:08 | GD | A | 250 | 14 | 200 | clear sky | - |
| 41 | 15 Oct 2017 | 11:48-15:51 | GD | M | 120 | 13 | 190 | clear sky | - |