# Peer review of "In-situ airborne measurements of atmospheric and sea surface parameters related to offshore wind parks in the German Bight"

_Earth System Science Data, 2019_

## Referee Comment (RC1) · Anonymous Referee #1 · 7 Jan 2020

**Review of:** In-situ airborne measurements of atmospheric and sea surface parameters related to offshore wind parks in the German Bight

**Authors:** A. Lampert et al.

**General Comments:**
This paper provides a description of a relatively unique and comprehensive dataset collected using an aircraft platform that contains measurements to evaluate the impact of offshore wind turbines on atmospheric properties. I believe that the paper is very appropriate for ESSD and that the material, in general, is well described. I did have several comments on the material in the paper that may help to improve the usefulness of the paper to readers outside of the team that collected these measurements. These are included below.

**Specific Comments:**
- Line 74: In the conversion to static temperature, are the pressure and temperature measurements co-located? It may be helpful to have some photographs or diagrams of the payload configuration on the aircraft to better understand how everything is laid out.
- Line 102: How does this 1.2 K uncertainty vary with temperature? How linear is this relationship? Is the 20 C value listed the instrument temperature, the air temperature or the surface temperature? Also, what impact does the vertical structure of temperature between the sensor and surface have on the quality of the measurement?
- Line 107: Significantly more information could be provided on how surface deflection is calculated using aircraft attitude corrections.
- Lines 120-126: Given that the camera images are not publicly available and that they are not included with the main dataset described here, does it make sense to provide more than one sentence on them? I'm not sure that the current configuration aligns with ESSD policies about data availability (not that I disagree with not making the imagery public).
- Lines 128-137: It would be very useful to show some statistics on the regions sampled (e.g. distributions of flight altitudes, distributions of distance from a known shore point, distributions of distance from known wind-farm points).
- Figure 2: What are the flight tracks that end abruptly at the coastline? There are several clusters that clearly are going into/out of an airport, but then there are also several singular lines that don't seem to go anywhere.
- Lines 130-131: It might be useful to show a wind rose with these directions on one or both of the maps. It may also be useful to show the extent of German-controlled airspace.
- Section 4.2: Without going into evaluation/analysis of the data, it might be useful to show an example dataset from one of these CROSS flights. One figure that illustrates the flight pattern followed and some of the structure that might be observed in the key quantities measured might be insightful for the reader.
- Section 5: For all of these quantities, it might be interesting to pick a few altitudes (e.g. 100, 200, 500 m) and come up with distributions of the mean quantities at these altitudes to plot. This will provide insight into the heterogeneity of the conditions sampled (or lack thereof). One could also imagine looking at scatter plots comparing these mean values (e.g. mean 50 m temperature vs. stability or similar). As an example, instead of showing the mean temperature profile and the range of values around that mean, perhaps it would be insightful to include the distribution of some "stability statistic" (e.g. LTS, etc.). Also,

while the mean profiles are interesting, maybe it would also be interesting to show a time-height plot of all of these profiles to demonstrate when the flights took place and whether there were clusters of flights that had similar conditions.

- Line 185: What is "cut-in speed"?
- Section 5.3: Here again, it might be more insightful to look at some statistic (e.g. ratio of wind at 50 m to wind at 100 m) as a function of other variables and/or location.
- Line 189: Am I understanding correctly that these profiles include all data, and aren't necessarily at a single location? This makes these very difficult to interpret.
- Lines 191-192: This could be a very interesting 3-component plot (e.g. a scatter plot of wind direction vs. strength, color-coded by stability).
- Lines 195-196: For unstable conditions, wouldn't you expect that mixing would result in an adiabatic profile that has an increasing RH with height?
- Line 196: Is the increasing humidity with altitude for stable profiles the result of layered advection impacting temperature, or are there moisture plumes being advected? Or something else?
- Section 6: I would have liked to have seen a bit more information in this conclusion. Some discussion on other complementary datasets, how these are expected to be used, etc. could be useful. Also, surprisingly, there was no mention of anything going wrong/not working/etc. during the campaign, which would be very remarkable for a campaign of this extent. Did anything go wrong that the reader should know about?

**Technical Corrections:**
- None at this time. The grammar, while sometimes different than I would have personally used, is perfectly suitable and readable. Perhaps the editorial team finds reasons to reword/correct, but I did not come across anything that required being corrected/changed.

---

## Referee Comment (RC2) · Christof Lüpkes (Referee) · 7 Jan 2020

General In this paper an aircraft campaign is described, which has been carried out in 2016 over North Sea in the environment of wind parks. The instrumentation, flight patterns and data sets are explained, which have been made publicly available in the world data center PANGAEA. The paper is generally well written and the unique data sets will be helpful to better understand the impact of wind turbines on atmospheric processes and the water surface. I recommend publication after some revisions mentioned below have been carried out.

Revisions 1) I understand that the goal of the authors is to just describe the data sets

[Figure]

and to leave the interpretation to later work. However, it might be possible to show in this paper at least one example illustrating the impact of the wind parks, which is missing in the current version. This would attract more readers.

2) Figures 5 and 6 show an unusual structure of the atmospheric boundary layer (ABL). The authors interpreted the peak at 500 m as the ABL top. Usually, a capping inversion is found at the top but here, at least in the average profile, the contrary (unstable stratification) is found. Even more pronounced is such a behavior at 950 m. This situation needs to be explained. I recommend showing two or three examples of individual (thus non-averaged) profiles in addition, since this might help to understand the mean profiles.

3) Measurements below 50 m height are obtained during take-off or landing and thus over land. So, it has nothing to do with the situation over sea and therefore, I would either mark these results with another color or skip them since this might lead to mis-interpretation. The average surface temperature is however interesting, but again, the land part should be excluded from the average.

4) The language is ok in general, but I found some misprints and also some unclear sentences: line 9: it should be written ....data set has been shown already by ... line 27: submitted by Bär line 29: as well as for (skip 'a') line 30: WIPAFF has been... line 37: probably, the authors mean resolution is more than 1 m ( or larger than...) line 39: and in particular over the sea ice ... line 58: meteorological line 60: sentence with resolution is a repetition of the introduction. The corresponding sentence of the introduction could perhaps occur here (?) line 73: give value of kappa. Line 105: and to derive Line 112: explain sigma line 142: instead of behind write 'downstream' line 165: I do not really understand the double averaging procedure for mean values. Does the first average mean the application of a filter so that, e.g., 1 Hz data result? line 166: Not completely clear how to understand the minimum and maximum. At each height, the minimum and maximum values were determined from all available profiles together? Line 206: better write something like: data base to date, from which the

impact . . .. can be derived.

Figures: Figures 3 and 4: use text size for label size as in the other figures. Times cannot be read. Either skip them or mark them in a different manner. German headings should be skipped as well and information should be given in the captions. Explain scale for wind speed.

─────────────────────────

---

## Author Comment (AC1) · 17 Mar 2020

[essd, manuscript]copernicus

[1]AstridLampert    [1]KonradBärfuss    [2]AndreasPlatis    [3]SimonSiedersleben [4]BughsinDjath [5]BeatrizCañadillas [1]RobertHunger [1]RudolfHankers [1]Mark-Bitter [1]ThomasFeuerle [1]HelmutSchulz [1]ThomasRausch [1]MaikAngermann [1]AlexanderSchwithal [2]JensBange [4]JohannesSchulz-Stellenfleth [5]ThomasNeumann [3]StefanEmeis

[1]Institute of Flight Guidance, Technische Universität Braunschweig, Braunschweig,

[Figure]

Germany [2]Eberhard Karls University, Tübingen, Germany [3]Karlsruhe Institute of Technology, Garmisch-Partenkirchen, Germany [4]Helmholtz Center for Material and Coastal Research, Geesthacht, Germany [5]UL International, Oldenburg, Germany

Airborne measurements of offshore wind park wakes

A. Lampert et al.

Astrid Lampert (Astrid.Lampert@tu-braunschweig.de)

**Answers to the referees on the article "In-situ airborne measurements of atmospheric and sea surface parameters related to offshore wind parks in the German Bight"**

March 17, 2020

**1   Answers to referee 1**

Dear Reviewer,
We thank you for your kind comments and for taking the time to consider our submitted manuscript. We have incorporated your advice into the revised manuscript that will be resubmitted. Below you will find our direct responses to the comments in normal letters. The comments are given in italic. Our changes to the text are additionally presented in quotation marks.

*This paper provides a description of a relatively unique and comprehensive dataset collected using an aircraft platform that contains measurements to evaluate the impact of offshore wind turbines on atmospheric properties. I believe that the paper is very*

*appropriate for ESSD and that the material, in general, is well described. I did have several comments on the material in the paper that may help to improve the usefulness of the paper to readers outside of the team that collected these measurements. These are included below.*

The authors would like to thank the referee for this positive overall judgement.

*Specific Comments:*

*- Line 74: In the conversion to static temperature, are the pressure and temperature measurements co-located? It may be helpful to have some photographs or diagrams of the payload configuration on the aircraft to better understand how everything is laid out.*

We added a figue showing the nose boom with the sensors and added in the text: "the central sensor package is contained in the nose boom (Fig. 1)."

*- Line 102: How does this 1.2 K uncertainty vary with temperature? How linear is this relationship? Is the 20 C value listed the instrument temperature, the air temperature or the surface temperature? Also, what impact does the vertical structure of temperature between the sensor and surface have on the quality of the measurement?*

We changed the text to: " It has an accuracy of $\pm 1.2$ K at $20^\circ$ C surface temperature and a temporal resolution of 20 Hz. If no clouds are between the sensor and the surface, the surface temperature measurements are not influenced by the atmospheric temperature or humidity distribution."

*- Line 107: Significantly more information could be provided on how surface deflection is calculated using aircraft attitude corrections.*

We added in the text: "From the point measurements in the scanner's coordinate system $(v)_{x_{body}} \, v_{y_{body}} \, v_{z_{body}}$, aircraft attitude corrections using Eulerian angles $\Psi, \Theta, \Phi$ are applied to rotate aircraft body fixed coordinates into the geodetic coordinate system

(positive directions East, North, Up), which then are geolocated by applying the aircraft position $(p)_{x_{body}} p_{y_{body}} p_{z_{body}}$ in the following manner:

$$(p)_{x_{geo}} p_{y_{geo}} p_{z_{geo}} = R_{geo}^{body}(-\Psi, -\Theta, -\Phi)(v)_{x_{body}} v_{y_{body}} v_{z_{body}} + (p)_{x_{body}} p_{y_{body}} p_{z_{body}} \quad (1)$$

Subsequently, the surface deflection $\eta$ is calculated out of the georeferenced point cloud using mean sea level."

*- Lines 120-126: Given that the camera images are not publicly available and that they are not included with the main dataset described here, does it make sense to provide more than one sentence on them? I'm not sure that the current configuration aligns with ESSD policies about data availability (not that I disagree with not making the imagery public).*
The authors would like to present the whole measurement system of the campaign. If the information about the camera images is not available in the text, the readers will not know about these additional data. So we would like to leave the section in the text, together with the information in the data availability section.

*- Lines 128-137: It would be very useful to show some statistics on the regions sampled (e.g. distributions of flight altitudes, distributions of distance from a known shore point, distributions of distance from known wind-farm points).*
We added a column with the main flight altitude in the tables. We think that the map is the most useful illustration of the locations of the flight.

*- Figure 2: What are the flight tracks that end abruptly at the coastline? There are several clusters that clearly are going into/out of an airport, but then there are also several singular lines that don't seem to go anywhere.*
We added in the figure caption: "Flight tracks end when the data acquisition was shut

down."

*- Lines 130-131: It might be useful to show a wind rose with these directions on one
or both of the maps.*
We think that the wind rose is too much information for the map. Anyway measure-
ments were not restricted to the indicated wind directions. This was the general idea,
but we have several examples (e.g. Flight 5) where we chose the wind park differently
due to other reasons (different flight pattern, etc.). We changed the text to: "Generally,
flights were performed downwind of Amrumbank West for a wind direction sector of
80° to 200°, and downwind of Godewind for a sector from 160° to 350°. However,
there are exceptions for particular reasons (e.g. during Flight 5 for consecutively
probing the wakes of both wind parks, Flight 6 for investigating the changes of the
wind field above the wind park)."

*It may also be useful to show the extent of German-controlled airspace.*
We included the permitted airspace in the figure.

*- Section 4.2: Without going into evaluation/analysis of the data, it might be useful to
show an example dataset from one of these CROSS flights. One figure that illustrates
the flight pattern followed and some of the structure that might be observed in the key
quantities measured might be insightful for the reader.*
We added references for each flight pattern in the sections. Further, we added an
example of the coastal effect.

*- Section 5: For all of these quantities, it might be interesting to pick a few altitudes
(e.g. 100, 200, 500 m) and come up with distributions of the mean quantities at
these altitudes to plot. This will provide insight into the heterogeneity of the conditions*

*sampled (or lack thereof). One could also imagine looking at scatter plots comparing these mean values (e.g. mean 50 m temperature vs. stability or similar).*

According to the analysis shown in other publications based on the data set (Platis et al., 2018; Siedersleben et al., 2018a,b; Platis et al., 2019a; Cañadillas et al., 2019; Siedersleben et al., 2019), the parameters strongly depend on each other. E.g. stable conditions are associated with flow from land if the land surface is warmer than the sea surface. Therefore, we think that scatter plots and statistics that are not taking into account the meteorological situation will not be useful for further analyses.

*As an example, instead of showing the mean temperature profile and the range of values around that mean, perhaps it would be insightful to include the distribution of some "stability statistic" (e.g. LTS, etc.). Also, while the mean profiles are interesting, maybe it would also be interesting to show a time-height plot of all of these profiles to demonstrate when the flights took place and whether there were clusters of flights that had similar conditions.*

The interaction of the atmospheric boundary layer with wind parks and coastal effects is very complex, and scatter plots or time series alone do not take into account the interactions. Results of detailed scientific analyses can be found in Platis et al. (2018); Siedersleben et al. (2018a,b); Platis et al. (2019a); Cañadillas et al. (2019); Siedersleben et al. (2019). As an example, we now include Fig. **??**, which shows the development of the wind speed at the altitude 120 m depending on the fetch length for all profiles (the distance from the coast along the wind direction). No systematic behaviour is obvious. The large scatter indicates that more parameters are necessary to understand the development of wind speed with fetch length. One of them is stability.

*- Line 185: What is "cut-in speed"?*

We changed the text to: "The typical cut-in speed at which offshore wind turbines start producing power is around $3\,\mathrm{m\,s^{-1}}$."

*- Section 5.3: Here again, it might be more insightful to look at some statistic (e.g. ratio of wind at 50 m to wind at 100 m) as a function of other variables and/or location.* As discussed above, we would prefer to leave statistical information to dedicated studies, as the interactions require complex analyses of different parameters. We added an example of the modification of the wind speed by coastal effects:
"The profiles of temperature and wind speed are modified by coastal effects (e.g. Dörenkämper et al. (2015); van der Laan et al. (2017)). During the WIPAFF campaign, climb and descent flights were performed on the way to the wind park measurement area and back. As an example of current and future research, the modification of the wind by coastal effects was investigated. Figure 2 shows the difference of the wind speed at hub height (120 m) for each profile minus the wind spee at hub height obtained during the profile closest to the coast. The fetch length is defined as the mean length that the air travelled above open water along the wind direction. Only flights are included where it was possible to determine the fetch length (not from North and West, as the distances to the next coast lines are too large). There is a large scatter in the data. Figure 3 shows the same data points. However, they are grouped by wind direction. There is still large scatter in the different data sets. However, depending on wind direction, the wind speed either increases or decreases with fetch length. This shows that there are more parameters required to explain the modification of wind speed besides the fetch length. A weakness of this analysis is that the profiles were not obtained along the mean wind direction. So air masses do not have the same origin, and besides the fetch length, variability along the coast line influences the results. More investigation is required to understand and parameterize the coastal effect."

*- Line 189: Am I understanding correctly that these profiles include all data, and aren't necessarily at a single location? This makes these very difficult to interpret.*
Yes, all available data of the vertical profiles are included, irrespective of the location.

The figures intend to provide a rough overview of the meteorological conditions encountered during the campaigns. However, other ways of presentation (e.g. scatter plots) seem even more difficult to interpret, see above.

*- Lines 191-192: This could be a very interesting 3-component plot (e.g. a scatter plot of wind direction vs. strength, color-coded by stability).*
As described above, we included an example of the coastal effect. Fig. **??** shows the same information as Fig. **??**, with additional information on the wind direction. The linear regressions are very different for each wind sector than for all data points. However, the scatter of the data points is still very large.

*- Lines 195-196: For unstable conditions, wouldn't you expect that mixing would result in an adiabatic profile that has an increasing RH with height?*
We agree with the referee, and changed the sentence to: "For unstable conditions, an enhanced water vapour mixing ratio directly above the water surface was present. "

*- Line 196: Is the increasing humidity with altitude for stable profiles the result of lay-ered advection impacting temperature, or are there moisture plumes being advected? Or something else?*
We changed the sentence to: "For stable conditions, humidity was often increased at higher altitudes, which in most cases is most likely caused by advection of air masses with higher water vapour mixing ratio."

*- Section 6: I would have liked to have seen a bit more information in this conclusion. Some discussion on other complementary datasets, how these are expected to be used, etc. could be useful.*
We added in the conclusion section: "The unique data has been the base for different

studies, proving for the first time directly the horizontal extension of wakes downwind of offshore wind parks (Platis et al., 2018), quantifying the recovery of wind speed in dependence of stability (Cañadillas et al., 2019; Platis et al., 2019a), and for the validation of the WRF mesoscale model (Siedersleben et al., 2018a,b, 2019), which can then be used for larger scales and scenario calculations."

*Also, surprisingly, there was no mention of anything going wrong/not working/etc. during the campaign, which would be very remarkable for a campaign of this extent. Did anything go wrong that the reader should know about?*
We changed the text to: " Altogether, 41 measurement flights were conducted during different seasons, wind direction, wind speed and stability. An overview of the flights performed during WIPAFF and meteorological conditions is shown in Tab. 1 and 2. A map with all flight paths flown during WIPAFF is provided in Fig. 3. During the flights, no instrument failures occurred. Only during one flight, the data acquisition had to be re-started (Flight 35)."

*Technical Corrections:*
*- None at this time. The grammar, while sometimes different than I would have personally used, is perfectly suitable and readable. Perhaps the editorial team finds reasons to reword/correct, but I did not come across anything that required being corrected/changed.*
Thank you!

[revised manuscript text omitted]

---

## Author Comment (AC2) · 17 Mar 2020

11

[Figure]

**Answers to the referees on the article "In-situ airborne measurements of atmospheric and sea surface parameters related to offshore wind parks in the German Bight"**

March 17, 2020

**1 Answers to referee 2**

The authors would like to thank Christof Lüpkes for the interesting comments.

In the following, each comment is addressed. The comments are given in italic, and the answers are given in normal letters. Changes to the text are provided in quotation marks.

*General*
*In this paper an aircraft campaign is described, which has been carried out in 2016 over North Sea in the environment of wind parks. The instrumentation, flight patterns*

*and data sets are explained, which have been made publicly available in the world data center PANGAEA. The paper is generally well written and the unique data sets will be helpful to better understand the impact of wind turbines on atmospheric processes and the water surface. I recommend publication after some revisions mentioned below have been carried out.*

We would like to thank the referee for the positive feedback.

*Revisions 1) I understand that the goal of the authors is to just describe the data sets and to leave the interpretation to later work. However, it might be possible to show in this paper at least one example illustrating the impact of the wind parks, which is missing in the current version. This would attract more readers.*

We included an example of changes of the wind when crossing the coast line. The analyses show that the development of the wind profiles from the coast to the open sea are difficult to understand and require more detailed analyses.

*2) Figures 5 and 6 show an unusual structure of the atmospheric boundary layer (ABL). The authors interpreted the peak at 500 m as the ABL top. Usually, a capping inversion is found at the top but here, at least in the average profile, the contrary (unstable stratification) is found. Even more pronounced is such a behavior at 950 m. This situation needs to be explained. I recommend showing two or three examples of individual (thus non-averaged) profiles in addition, since this might help to understand the mean profiles.*

We included individual profiles of Flight 7 (stable), Flight 15 (unstable), Flight 31 (stable) in the figures of temperature, potential temperature, water vapour mixing ratio and wind speed.

We added in the text:

"Below 60 m, data are only available during take-off and landing. Therefore, the temperature inversion below 60 m is not a typical feature above the North Sea, and

therefore provided as dotted line. The averaged and maximum temperature profiles show a sudden decrease at an altitude of around 500 m and around 950 m. This is probably an artifact from the averaging method, and it is not visible in individual temperature profiles."

"In the mean profile of the potential temperature, a clear increase is observed for the altitude interval 60 to 100 m. Also up to the altitude of 200 m, in the range of the rotor blades, an overall small increase of potential temperature with height is observed. The decrease of average and maximum potential temperatures with height at around 500 m and 950 m are probably artifacts form the averaging method and are not visible in the profiles of individual flights."

*3) Measurements below 50 m height are obtained during take-off or landing and thus over land. So, it has nothing to do with the situation over sea and therefore, I would either mark these results with another color or skip them since this might lead to misinterpretation. The average surface temperature is however interesting, but again, the land part should be excluded from the average.*
We now provide the profiles below 60 m, which were obtained above land during take-off and landing, in dotted lines.

*4) The language is ok in general, but I found some misprints and also some unclear sentences:*
*line 9: it should be written ...data set has been shown already by...*
We changed as suggested.

*line 27: submitted by Baerfuss*
We did as suggested.

*line 29: as well as for (skip 'a')*
Done.

*line 30: WIPAFF has been...*
Done.

*line 37: probably, the authors mean resolution is more than 1 m ( or larger than...)*
We changed the text to "the spatial resolution of the measurements is better than 1 m"

*line 39: and in particular over the sea ice...*
We changed as suggested.

*line 58: meteorological*
We corrected the spelling.

*line 60: sentence with resolution is a repetition of the introduction. The corresponding sentence of the introduction could perhaps occur here (?)*
We moved the sentence from the earlier section here as suggested.

*line 73: give value of kappa.*
We changed the text to: "$\kappa$ is the heat capacity ratio with a value of 1.4."
*Line 105: and to derive*

We changed the text to: "The scanning laser system VZ-1000 of Riegl, Austria, is deployed to record the relative sea surface deflection and to derive parameters like the significant wave height."

*Line 112: explain sigma*
We changed the text to: "since this simple calculus only depends on the standard deviation $\sigma_\eta$ in sea surface deflection"

*line 142: instead of behind write 'downstream'*
We changed as suggested.

*line 165: I do not really understand the double averaging procedure for mean values. Does the first average mean the application of a filter so that, e.g., 1 Hz data result?*
To clarify, we changed the text to: "During each flight, different vertical profiles were obtained. First, a mean profile for each flight was calculated. Then all 41 profiles from the 41 flights were averaged again. The minimum and maximum values are combined from the 41 mean profiles."

*line 166: Not completely clear how to understand the minimum and maximum. At each height, the minimum and maximum values were determined from all available profiles together?*
We changed the text to: "For each height, the minimum and maximum values were determined from the 41 profiles representing each one particular flight."

*Line 206: better write something like: data base to date, from which the impact.... can be derived.*
We changed the text to: "The WIPAFF flights are the only available data base to date, from which the impact of long-range wakes can be derived systematically and independent of infrastructural constraints like the location of masts."

*Figures: Figures 3 and 4: use text size for label size as in the other figures. Times*

*cannot be read. Either skip them or mark them in a different manner. German headings should be skipped as well and information should be given in the captions. Explain scale for wind speed.*

We now use larger label size and omitted the times and the German headings. The colours indicating the altitude and the indications for wind direction are explained in the caption: "The colours show the flight altitude. The wind barbs indicate wind direction and a first idea of wind speed, which is proportional to the length of the wind barbs."